# Individual and community level determinants of short birth interval in Ethiopia: A multilevel analysis

Desalegn Markos Shifti [1,2]*, Catherine Chojenta[2], Elizabeth G. Holliday[3], Deborah Loxton[2]

**1** Saint Paul's Hospital Millennium Medical College, Addis Ababa, Ethiopia, **2** Priority Research Centre for Generational Health and Ageing, School of Medicine and Public Health, University of Newcastle, Newcastle, New South Wales, Australia, **3** Centre for Clinical Epidemiology and Biostatistics, School of Medicine and Public Health, University of Newcastle, Newcastle, New South Wales, Australia

* desalegnmarkos@gmail.com, desalegnmarkos.shifti@uon.edu.au

**Data Availability Statement:** All data are available from the DHS Program database (https://dhsprogram.com/data/available-datasets.cfm). Any researcher can request for access to the data without any restriction.

## Abstract

### Background

The World Health Organization recommends a minimum of 33 months between two consecutive live births to reduce the risk of adverse maternal and child health outcomes. However, determinants of short birth interval have not been well understood in Ethiopia.

### Objective

The aim of this study was to assess individual- and community-level determinants of short birth interval among women in Ethiopia.

### Methods

A detailed analysis of the 2016 Ethiopian Demographic and Health Survey data was performed. A total of 8,448 women were included in the analysis. A two-level multilevel logistic regression analysis was used to identify associated individual- and community-level factors and estimate between-community variance.

### Results

At the individual-level, women aged between 20 and 24 years at first marriage (AOR = 1.37; 95% CI: 1.18–1.60), women aged between 25 and 29 years at first marriage (AOR = 1.65; 95% CI: 1.20–2.25), having a husband who attended higher education (AOR = 1.32; 95% CI: 1.01–1.73), being unemployed (AOR = 1.16; 95% CI: 1.03–1.31), having an unemployed husband (AOR = 1.23; 95% CI: 1.04–1.45), being in the poorest wealth quintile (AOR = 1.82; 95% CI: 1.39–2.39), being in the poorer wealth quintile (AOR = 1.58; 95% CI: 1.21–2.06), being in the middle wealth quintile (AOR = 1.61; 95% CI: 1.24–2.10), being in the richer wealth quintile (AOR = 1.54; 95% CI: 1.19–2.00), increased total number of children born before the index child (AOR = 1.07; 95% CI: 1.03–1.10) and death of the preceding child (AOR = 1.97; 95% CI: 1.59–2.45) were associated with increased odds of short birth

**Funding:** The authors received no specific funding for this work.

**Competing interests:** The authors have declared that no competing interests exist.

interval. At the community-level, living in a pastoralist region (AOR = 2.01; 95% CI: 1.68–2.39), being a city dweller (AOR = 1.75; 95% CI: 1.38–2.22), high community-level female illiteracy (AOR = 1.23; 95% CI: 1.05–1.45) and increased distance to health facilities (AOR = 1.32; 95% CI: 1.11–1.56) were associated with higher odds of experiencing short birth interval. Random effects showed significant variation in short birth interval between communities.

## Conclusion

Determinants of short birth interval are varied and complex. Multifaceted intervention approaches supported by policy initiatives are required to prevent short birth interval.

## Introduction

A preceding birth interval is defined as the amount of time between the birth of the child under study (index child) and the immediately preceding birth [1, 2]. It has received increased attention in demography and public health research because of its implications for fertility and maternal and child health [3].

To reduce the risk of adverse maternal and child health outcomes, the most recent World Health Organization (WHO) recommendation for a healthy pregnancy interval is at least two years (24 months) or a birth-to-birth interval of 33 months, assuming nine months gestation [4]. Short birth interval is associated with increased risk of adverse birth outcomes, such as pre-term birth [5, 6], low birth weight [5, 6], small size for gestational age [5], congenital anomalies [7, 8], autism [9], and infant mortality [5, 10]. It is also associated with a higher risk of maternal morbidities, including preeclampsia, high blood pressure, and premature rupture of membranes [11].

Worldwide, birth interval practices differ widely [12]. Particularly, women in developing countries often have shorter birth intervals than they would personally prefer [13]. For instance, in sub-Saharan Africa, mothers would prefer a median birth interval of 38.9 months, which is more than 6.2 months longer than their actual median birth interval [14]. Similarly, evidence from Demographic Health Survey data (DHS) of 52 different countries showed that more than two-thirds of births occurred at durations of less than 30 months since the previous birth [15].

A variety of factors influences a woman's birth interval, including her personal characteristics and the health status of her previous child [12]. Traditional practices, particularly breastfeeding, postpartum abstinence and cultural norms also affect birth spacing [12]. Other factors affecting birth interval are rooted in social and cultural norms, including reproductive histories and behaviors of individual women, utilization of reproductive health services and other background factors [16].

In Ethiopia, the prevalence of short birth interval ranges from 21.7% (national estimate) [17] to 57.6% (district level estimate) [18]. In addition, there were inconsistencies among studies in identifying determinants of short birth interval, such as wealth index [18–20], maternal age [19, 21], and the number of children [19, 21]. Moreover, previous studies [18–21] mainly focused on individual-level determinants of short birth interval. However, some of the individual-level factors such as marital status, maternal age at marriage, maternal age at birth, death of preceding child, sex of preceding child, exposure to media, and distance to a health facility have not been well studied in Ethiopia. Furthermore, community-level factors such as regions

where women reside, community-level female literacy, and community-level poverty have not yet been investigated. However, human population has a complex community structure and its characteristics can also affect an individual's health behavior, including birth interval practices. Hence, it is important to assess the effects of individual- and community-level factors together to understand the complex nature of short birth interval, which can then inform national-level policy direction as well as individual- and community-based interventions, such as mass campaigns and dissemination of health education messages.

Taking into account changing demographics, such as high fertility [17] in Ethiopia, the increasing proportion of short birth intervals [18, 20] with a marked variation among regions ranging from 10.6% in Amhara region to 45.9% in Somali region [17], our objective was to identify individual- and community-level determinants of short birth interval in Ethiopia. The findings of this study will provide pertinent information for policy makers, program planners and other stakeholders, which in turn may help to design and implement appropriate interventions at different levels to prevent short birth interval and improve women's and children's health in general.

## Methods

### Study area and setting

An in-depth secondary data analysis was conducted using the 2016 Ethiopian Demographic and Health Survey (EDHS) data. The EDHS is carried out every five years to provide health and health-related indicators at the national and regional levels in Ethiopia. The 2016 EDHS is the fourth national survey conducted in all parts of Ethiopia (in nine regional states [Tigray, Afar, Amhara, Oromia, Somali, Benishangul-Gumuz, Southern Nations, Nationalities and Peoples' Region (SNNPR), Gambella, and Harari] and two administrative cities [Addis Ababa and Dire Dawa]). Administratively, regions in Ethiopia are divided into Zones, and Zones into administrative units called *Woredas*. Each *Woreda* is further subdivided into the lowest administrative unit, called *Kebeles*. During the 2007 census, each *Kebele* was subdivided into census enumeration areas (EAs), which were convenient for the implementation of the census.

### Study design and sampling

The 2016 EDHS was cross-sectional by design. It employed two-stage stratified cluster sampling based on a sampling frame of the 2007 Population and Housing Censuses in which EAs were the sampling units for the first stage and households for the second stage (Fig 1). The detailed sampling procedure is presented in the full EDHS report [17]. The current study included individual-level data for 8,448 women who had at least two live births during the five years preceding 2016 as well as community characteristics of 640 clusters. Women who had never been married were excluded from the analysis, since women who have multiple births out of wedlock are unlikely to plan their births in the same way as married women. Data collection took place from 18 January 2016 to 27 June 2016.

### Outcome variable

The outcome variable for this study was short birth interval. Short birth interval was defined as an interval less than 33 months between two successive live births (33 months = 24 months of birth to conception period + 9 months duration of pregnancy). Preceding birth intervals exceeding 33 months were defined as non-short birth interval, which is consistent with the WHO recommendations [4].

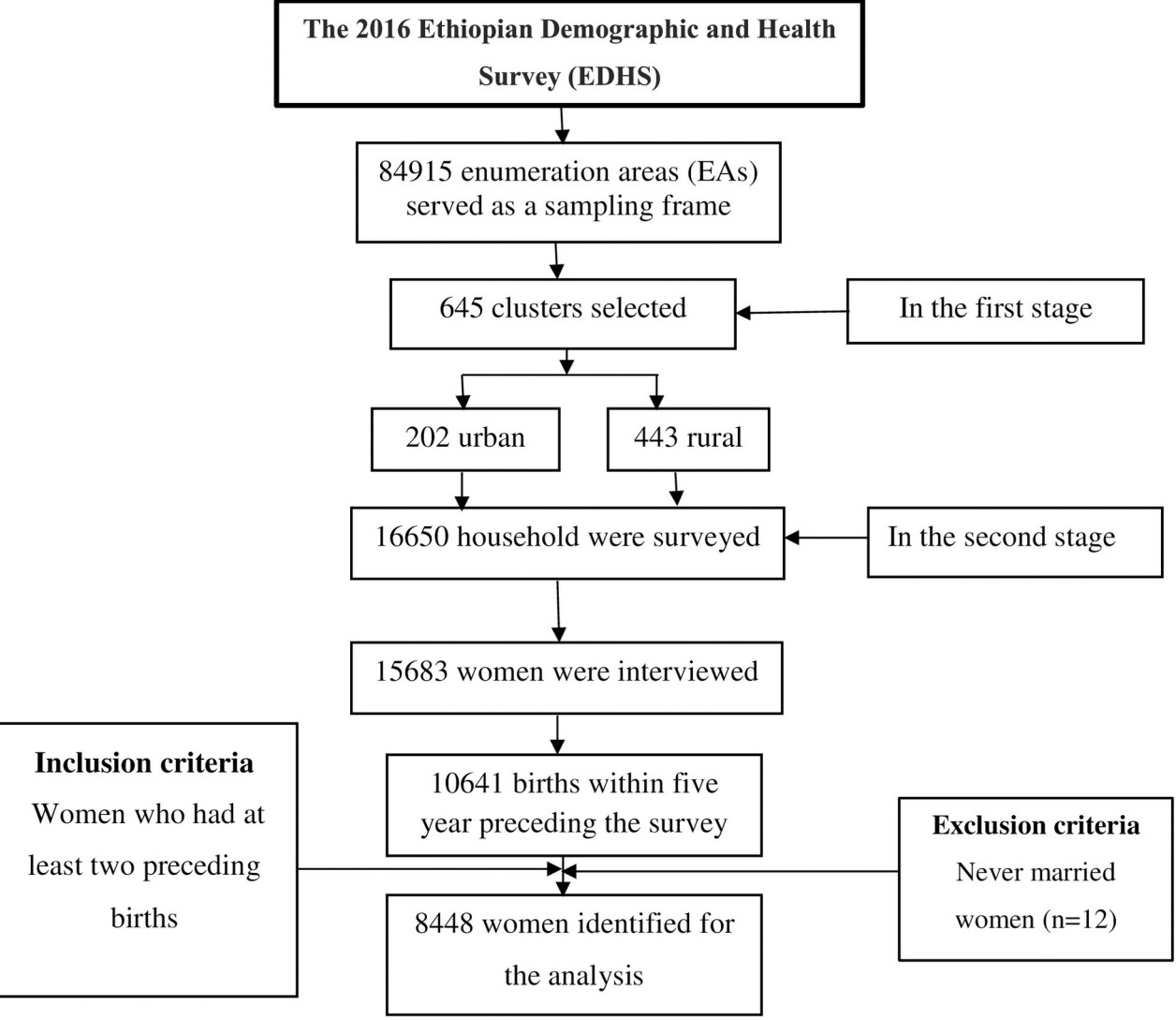

**Fig 1. The flowchart for the sampling and data extraction procedure, EDHS 2016** *EDHS*, **Ethiopian Demographic and Health Survey.**

## Exposure variables

The socio-ecological model provides the theoretical foundation for selecting the exposure variables. The model asserts that human health behavior, the experience of short birth interval in the current study, is influenced at multiple levels within the social and physical environment, such as the individual-, household-, community- and societal/policy-level factors [22–24]. The framework also assumes that interactions between individuals and their environment are reciprocal, implying that individuals are influenced by their environment and the environment is influenced by individuals [25]. For instance, an individual woman's literacy status is affected by community-level female literacy and vice versa. These variables, in turn, may independently affect women's experience of short birth interval. An increased understanding of the impact of individual- and community-level factors on the risk of short birth interval can inform policy makers and health programmers by providing insight into how to prevent the risk of short birth interval among individuals as well as groups of women in a community. Assessing the

impact of societal/policy-level factors on the risk of short birth interval, however, is beyond the scope of the current study.

Details of the definition and coding of the individual- and community-level factors included in the current study are presented online (see S1 Table). Variables were selected based on reviewed literature [2, 18–21, 26–31]. Individual factors included maternal age at first marriage, maternal age at birth of the preceding child, marital status, polygyny status, maternal educational level, husbands'/partners' education level, maternal occupation, husbands'/partners' occupation, wealth quintile, sex of preceding child, total number of children born before the index child, survival status of preceding child, exposure to media (television, radio and newspaper; each separately) and distance to health facility. Separated, divorced, and widowed women were considered in the analysis. Since data regarding birth interval of women were collected from their most recent childbirth in the five years preceding the survey, women may have become widowed or divorced or separated before the time of the survey but after they already became eligible to provide birth interval information. Community-level factors included place of residence, contextual regions, community-level female illiteracy, community-level poverty, and community-level distance to a health facility. The 11 regions of Ethiopia are essentially delineated for administrative purposes, and in this study they were categorized into three contextual regions: pastoralist, agrarian, and city (which were defined on the basis of the cultural and socio-economic backgrounds of their populations) [32]. Except for place of residence and geographic region, the EDHS did not capture variables that describe community characteristics. Therefore, additional community-level characteristics were constructed by aggregating the individual-level characteristics within their clusters. Aggregate values were classified as high or low based on the distribution in each community. The cut-point was defined as the mean if the aggregate variable was normally distributed and the median if not. For example, community-level poverty refers to the proportion of poor households (two lowest wealth quintiles) in the community (cut-off at median proportion). Community-level factors describe groups of populations living in similar settings.

The assumption of independence of observation has been taken as a basis to determine whether variables were analysed at the individual or community-level. If the observations at the individual-level are independent, variables were treated as individual-level factors. Whereas if the observations were clustered into higher levels of units or if several women had shared features (e.g., place of residence and contextual region, that could have the same effect on short birth interval among women in that locality), then the variables were analysed at the community-level.

## Data analysis

**Descriptive analysis.** Due to the non-proportional allocation of the sample to different regions as well as urban and rural areas, and the possible differences in response rates, as per the recommendation of the EDHS, proportions and frequencies were estimated after applying sample weights to the data to adjust for disproportionate sampling and non-responses. A detailed explanation of the weighting procedure can be found in the EDHS report [17]. To examine the crude association between the individual- and community-level factors separately with short birth interval, p-values were calculated using Pearson's chi-squared test. A p-value of less than 0.05 was set for statistical significance of an association.

**Multivariable multilevel analysis.** The 2016 EDHS employed multistage cluster sampling and data are hierarchical (i.e., mothers are nested within households, and households are nested within clusters). As a result of the sampling approach, mothers within the same cluster may be more similar to each other than mothers in the rest of the country. To account for this

clustering, two-stage multivariable multilevel logistic regression analysis was used to estimate the effects of individual- and community-level determinants on short birth interval and to esti- mate the between-cluster variability in the odds of short birth interval. For the multilevel logistic regression analysis, the Stata syntax *xtmelogit* was employed [33]. Four models were constructed in this analysis. Model I (the empty model) was fitted without explanatory variables to estimate random variation in the intercept and the intra-cluster correlation coefficient (ICC; i.e., to eval- uate the extent of the cluster variation in short birth interval). Model II was constructed to examine the effects of individual-level characteristics, while Model III was fitted to assess the effects of community-level characteristics. Finally, model IV adjusted for the individual- and community-level variables simultaneously. The fixed effect sizes of individual- and community- level factors on short birth interval were expressed using the adjusted odds ratios (AOR) with 95% confidence interval (CI). A p-value of $<0.05$ was used to declare statistical significance.

Total number children born before the index child was considered as a continuous variable. This was done after checking for the linearity assumption with the log-odds of short birth interval, which is a binary response variable. Multicollinearity was also checked among the exposure variables using the variance inflation factor (VIF). If the values of VIF were lower than 10, then the collinearity problem was considered to be unlikely. The VIF for birth order was 15.80 and for the total number of children born before the index child was 15.77, which indicates the presence of collinearity. Therefore, we removed the variable birth order from the model and the VIF became less than 2 for each variable included in the model (see S2 Table).

The random effects were measured by the intra-class correlation coefficient (ICC), median odds ratio (MOR) and proportional change in variance (PCV). The ICC was calculated to evalu- ate whether the variation in short birth interval is primarily within or between communities [34]. This is because individuals living in the same neighbourhood (community) may be more similar to each other than individuals living in other neighbourhoods as they share a number of eco- nomic, social, and other neighbourhood characteristics that may condition similar health status [35]. The ICC for two-level binary data was calculated using the following formula [34, 36]:

$$\text{ICC} = \frac{V_a}{V_a + \frac{\pi^2}{3}}$$

Where $V_a$ $(\tau^2)$ is community-level variance and $\frac{\pi^2}{3}$ is individual-level variance (VI) equal to 3.29. It takes a value between 0 and 1. A high ICC value indicates that neighbourhoods are important in understanding individual differences in short birth interval [37]. The MOR is defined as the median value of the odds ratio between the area at highest risk and the area at the lowest risk when randomly picking out two areas and it depends directly on the area-level variance [34]. It can be calculated using the following formula:

$$\text{MOR} = (\exp \ \sqrt{2 \text{ X } V_a \text{ X } 0.6745}) \approx \exp(0.95 \ \sqrt{V_A})$$

Where $V_a$ is the area-level variance, and 0.6745 is the 75[th] centile of the cumulative distribu- tion function of the normal distribution with mean 0 and variance 1. In this study, MOR shows the extent to which the individual probability of experiencing short birth interval is determined by the residential area [34]. The PCV is used to measure the total variation attrib- uted to individual-level factors and area-level factors in the multilevel model [34, 38]. It was calculated using the following mathematical equation:

$$\text{PCV} = \frac{V_{n-1} - V_{n-2}}{V_{n-1}}$$

**Table 1. Weighted proportion of short birth interval by individual-level factors, EDHS 2016.**

| Variables | Weighted proportion | | P value |
|---|---|---|---|
| | **Non-short birth interval** | **Short birth interval** | |
| **Maternal age at first marriage (years)** | | | <0.001 |
| ≤19 | 3499 (83.6) | 3324 (79.4) | <0.001 |
| 20–24 | 600 (12.7) | 697 (16.3) | |
| 25–29 | 116 (2.8) | 142 (3.6) | |
| 30+ | 46 (0.9) | 24 (0.7) | |
| **Maternal age at birth of the preceding child (years)** | | | |
| ≤19 | 712 (16.7) | 598 (13.8) | |
| 20–24 | 1383 (31.3) | 1391 (32.6) | |
| 25–29 | 1117 (26.6) | 1143 (27.9) | |
| 30–34 | 723 (16.9) | 706 (16.4) | |
| 35+ | 326 (8.5) | 349 (9.3) | |
| **Marital status** | | | |
| Separated/Divorced/Widowed | 227 (4.2) | 144 (2.9) | <0.001 |
| Married | 4034 (95.8) | 4043 (97.1) | |
| **Polygyny status (n = 8077)** | | | |
| Yes | 552 (11.4) | 774 (14.0) | <0.001 |
| No | 3467 (88.2) | 3257 (85.4) | |
| Don't know | 15 (0.4) | 12 (0.5) | |
| **Maternal educational level** | | | |
| No education | 2871 (71.9) | 3201 (76.4) | <0.001 |
| Primary | 1019 (22.9) | 779 (20.7) | |
| Secondary | 250 (3.4) | 131 (1.8) | |
| Higher | 121 (1.7) | 76 (1.1) | |
| **Husband's/partner's education level (n = 8077)** | | | |
| No education | 2023 (52.0) | 2308 (51.6) | <0.001 |
| Primary | 1341 (37.3) | 1210 (40.3) | |
| Secondary | 365 (6.2) | 279 (4.5) | |
| Higher | 281 (4.1) | 217 (2.6) | |
| Don't know | 24 (0.4) | 29 (1.0) | |
| **Maternal occupation** | | | |
| Not working | 2949 (71.1) | 3164 (75.1) | <0.001 |
| Working | 1312 (28.9) | 1023 (24.9) | |
| **Husband's/partner's occupation (n = 8077)** | | | |
| Not working | 350 (6.5) | 545 (9.5) | <0.001 |
| Working | 3639 (93.0) | 3466 (89.8) | |
| Don't know | 45 (0.5) | 32 (0.6) | |
| **Wealth quintiles** | | | |
| Poorest | 1348 (22.0) | 1999 (28.8) | <0.001 |
| Poorer | 759 (22.1) | 719 (24.8) | |
| Middle | 658 (21.0) | 545 (21.5) | |
| Richer | 589 (19.1) | 466 (16.6) | |
| Richest | 907 (15.8) | 458 (8.3) | |
| **Sex of preceding child** | | | |
| Male | 2162 (50.9) | 2169 (52.0) | 0.33 |
| Female | 2099 (49.1) | 2018 (48.0) | |

(*Continued*)

**Table 1.** (*Continued*)

| Variables | Weighted proportion | | P value |
|---|---|---|---|
| | **Non-short birth interval** | **Short birth interval** | |
| **Survival status of preceding child** | | | <0.001 |
| Yes | 4099 (96.0) | 3855 (92.7) | |
| No | 162 (4.0) | 332 (7.3) | |
| **Distance to a health facility** | | | <0.001 |
| Big problem | 2279 (58.0) | 2530 (67.3) | |
| Not big problem | 1982 (42.0) | 1657 (32.7) | |
| **Watched television** | | | <0.001 |
| Yes | 946 (18.9) | 538 (12.9) | |
| No | 3315 (81.1) | 3649 (87.1) | |
| **Listened to radio** | | | <0.001 |
| Yes | 1086 (26.6) | 734 (23.0) | |
| No | 3179 (73.4) | 3453 (77.0) | |
| **Read newspapers** | | | <0.001 |
| Yes | 296 (5.6) | 143 (3.8) | |
| No | 3965 (94.4) | 4044 (96.2) | |

Where $V_{n-1}$ is the neighbourhood variance in the empty model and $V_{n-2}$ is the neighbourhood variance in the subsequent model.

**Model fit statistics.** Akaike's information criterion (AIC) and Schwarz's Bayesian information criteria (BIC) were used to assess goodness of fit and inform the selection of nested models (individual- and community-level model). The AIC and BIC values were compared in successive models and the model with the lowest value was considered to be the best-fit model [33, 39]. Unlike statistical methods that employ hypothesis testing approaches such as deviance differences and log-likelihood ratio statistics, AIC and BIC penalize the deviance for a larger number of parameters (overparameterization) [39–41]. Thus, they provide more protection against overfitting the model to the data. In addition, area under receiver operating characteristics (AUC) curve was used to assess the predictive quality of each model [42].

Statistical analysis was performed using Stata version 14 statistical software *(StataCorp. Stata Statistical Software*: *Release 14. College Station, TX*: *StataCorp LP. 2015)*.

### Ethics statement

The 2016 EDHS was approved by the National Research Ethics Review Committee of Ethiopia and ICF Macro International. Permission from The DHS Program was obtained to use the 2016 EDHS data. This analysis was approved by the University of Newcastle Human Research Ethics Committee (H-2018-0332).

## Results

### Sociodemographic and other health related characteristics

Overall, 8,448 women at level 1 nested within 640 clusters at level 2 were included in the analysis. The proportion of short birth intervals based on individual- and contextual-level background characteristics of the study participants are shown in Table 1. The mean (standard deviation) age of women who participated in the study was 30.82 (6.1). Among women who experienced short birth interval, 79.4% were 19 years old or under at their first marriage,

76.4% were uneducated, 75.1% were not working, and 28.8% were from the poorest households.

As depicted in Table 2, the vast majority of women with short birth interval resided in rural areas (94.0%) and agrarian regions (87.8%).

The overall prevalence of short birth interval in Ethiopia was 45.8% (95% CI: 42.91–48.62).

## Determinants of short birth interval among women

**Multilevel analysis (fixed effect analysis).**  Table 3 presents the results of multivariable multilevel logistic regression analysis (fixed effects). In the full model adjusted for individual- and community-level factors, maternal age at first marriage, maternal educational level, maternal occupation, husbands'/partners' occupation, wealth quintile, total number of children born before the index child, death of the preceding child, contextual regions, community-level illiteracy, community-level poverty, and community-level increased distance to health facility were each associated with short birth interval among women.

**Individual-level factors.**  The odds of short birth interval among women aged between 20 and 24 years at first marriage were 37% higher (AOR = 1.37; 95% CI: 1.18–1.60) compared with women whose age at their first marriage was 19 years or under. Likewise, the odds of short birth interval among women aged between 25 and 29 years at first marriage were 65% higher (AOR = 1.65; 95% CI: 1.20–2.25) compared with women whose age at first marriage was 19 years or under. The odds of short birth interval among not working women was 16% higher (AOR = 1.16; 95% CI: 1.03–1.31) than for working women. The odds of short birth interval were 23% higher (AOR = 1.23; 95% CI: 1.04–1.45) among women whose husband/partner was not working compared with women who had a working husband/partner. On the other hand, the odds of short birth interval were 46% lower (AOR = 0.54; 95% CI: 0.32–0.90) among women who did not know their partner's working status than women whose husband/partner was working. The odds of short birth interval among women from the poorest households were 1.82 times higher (AOR = 1.82; 95% CI: 1.39–2.39), from poorer households 1.58 times higher (AOR = 1.58; 95% CI: 1.21–2.06), from middle income households 1.61 times

**Table 2. Weighted proportion of short birth interval by community-level factors, EDHS 2016.**

| Variables | Weighted proportion | | P value |
|---|---|---|---|
| | Non-short birth interval | Short birth interval | |
| **Place of residence** | | | |
| Urban | 818 (11.3) | 468 (6.0) | <0.001 |
| Rural | 3443 (88.7) | 3719 (94.0) | |
| **Contextual regions** | | | |
| Agrarian | 2310 (93.0) | 1629 (87.8) | <0.001 |
| Pastoralist | 1338 (4.4) | 2065 (10.7) | |
| City dweller | 613 (2.6) | 493 (1.5) | |
| **Distance to a health facility as a big problem** | | | |
| High | 2233 (59.0) | 2630 (72.5) | <0.001 |
| Low | 2028 (41.0) | 1557 (27.5) | |
| **Community-level poverty** | | | |
| High | 1902 (34.7) | 2564 (42.2) | <0.001 |
| Low | 2359 (65.3) | 1623 (57.8) | |
| **Community-level female literacy** | | | |
| High | 1861 (48.4) | 2357 (50.2) | <0.001 |
| Low | 2400 (51.6) | 1830 (49.8) | |

**Table 3. Individual- and community-level determinants of short birth interval in Ethiopia using multilevel logistic regression analysis, EDHS 2016.**

| Variables | Model 2[a] AOR (95% CI) | Model 3[b] AOR (95% CI) | Model 4[c] AOR (95% CI) |
|---|---|---|---|
| **Maternal age at first marriage (reference:≤19 years)** | | | |
| 20–24 | 1.37 (1.18, 1.60)*** | | 1.37 (1.18, 1.60)*** |
| 25–29 | 1.63 (1.19, 2.23)** | | 1.65 (1.20, 2.25)** |
| 30+ | 0.69 (0.38, 1.24) | | 0.68 (0.38, 1.21) |
| **Maternal age at birth of the preceding child (reference: ≤19)** | | | |
| 20–24 | 1.07 (0.91, 1.25) | | 1.09 (0.93, 1.27) |
| 25–29 | 0.89 (0.74, 1.08) | | 0.94 (0.78, 1.13) |
| 30–34 | 0.78 (0.62, 0.98)** | | 0.83 (0.66, 1.05) |
| 35+ | 0.82 (0.61, 1.10) | | 0.89 (0.66, 1.20) |
| **Polygyny status (reference: not polygyny)** | | | |
| Polygyny | 1.09 (0.95, 1.26) | | 1.01 (0.88, 1.17) |
| Don't know | 1.50 (0.62, 3.62) | | 1.62 (0.67, 3.90) |
| **Maternal educational level (reference: No education)** | | | |
| Primary | 0.99 (0.86, 1.13) | | 1.03 (0.89, 1.19) |
| Secondary | 0.81 (0.60, 1.09) | | 0.83 (0.61, 1.12) |
| Higher | 1.22 (0.80, 1.87) | | 1.24 (0.81, 1.90) |
| **Husband's/partner's education level (reference: No education)** | | | |
| Primary | 1.07 (0.95, 1.21) | | 1.12 (0.99, 1.2) |
| Secondary | 1.19 (0.96, 1.47) | | 1.16 (0.94, 1.44) |
| Higher | 1.36 (1.04, 1.78)** | | 1.32 (1.01, 1.73)** |
| Don't know | 1.38 (0.74, 2.55) | | 1.41 (0.76, 2.62) |
| **Maternal occupation (reference: working)** | | | |
| Not working | 1.17 (1.04, 1.32)** | | 1.16 (1.03, 1.31)** |
| **Husband's/partner's occupation (reference: working)** | | | |
| Not working | 1.30 (1.09, 1.53)** | | 1.23 (1.04, 1.45)** |
| Don't know | 0.55 (0.33, 0.92)** | | 0.54 (0.32, 0.90)** |
| **Wealth quintile (reference: richest)** | | | |
| Poorest | 2.60 (2.08, 3.26)*** | | 1.82 (1.39, 2.39)*** |
| Poorer | 1.86 (1.47, 2.34)*** | | 1.58 (1.21, 2.06)** |
| Middle | 1.77 (1.40, 2.24)*** | | 1.61 (1.24, 2.10)*** |
| Richer | 1.67 (1.33, 2.11)*** | | 1.54 (1.19, 2.00)** |
| **Sex of preceding child (reference: male)** | | | |
| **Female** | 0.96 (0.87, 1.05) | | 0.96 (0.87, 1.06) |
| **Total number of children born before the index child** | 1.07 (1.04, 1.11)*** | | 1.07 (1.03, 1.10)*** |
| **Survival of preceding child (reference: yes)** | | | |
| No | 2.01 (1.62, 2.50)*** | | 1.97 (1.59, 2.45)*** |
| **Watched television (reference: yes)** | | | |
| No | 1.09 (0.91, 1.30) | | 1.06 (0.88, 1.27) |
| **Listened to radio (reference: yes)** | | | |
| No | 1.16 (1.01, 2.34)** | | 1.14 (0.99, 1.31) |
| **Read newspapers (reference: yes)** | | | |
| No | 1.15 (0.88, 1.51) | | 1.10 (0.84, 1.44) |
| **Distance to a health facility to get medical help (reference: not big problem)** | | | |

*(Continued)*

**Table 3.** (Continued)

| Variables | Model 2[a] AOR (95% CI) | Model 3[b] AOR (95% CI) | Model 4[c] AOR (95% CI) |
|---|---|---|---|
| Big problem | 1.02 (0.91, 1.15) | | 0.94 (0.83, 1.06) |
| **Community-level factors** | | | |
| **Place of residence (reference: urban)** | | | |
| Rural | | 1.72 (1.36, 2.17)*** | 1.23 (0.93, 1.63) |
| **Contextual regions (reference: Agrarian)** | | | |
| Pastoralist | | 2.24 (1.88, 2.67)*** | 2.01 (1.68, 2.39)*** |
| City dweller | | 1.68 (1.32, 2.13)*** | 1.75 (1.38, 2.22)*** |
| **Community-level female illiteracy (reference: low)** | | | |
| High | | 1.29 (1.10, 1.48)** | 1.23 (1.05, 1.45)** |
| **Community-level poverty (reference: low)** | | | |
| High | | 1.23 (1.02, 1.48)** | 1.13 (0.93, 1.37) |
| **Community-level distance to the health facility as a big problem (reference: low)** | | | |
| High | | 1.38 (1.17, 1.62)*** | 1.32 (1.11, 1.56)** |

***P value <0.001

** P value <0.05

AOR = Adjusted Odds Ratio; CI = Confidence Interval

Model 2[a] is adjusted for individual-level factors

Model 3[b] is adjusted for community-level factors

Model 4[c] is the final model adjusted for individual- and community-level factors

N.B. Model 1 (Empty model) was fitted without determinant variables and is not included in this table (but is in Table 4).

higher (AOR = 1.61; 95% CI: 1.24–2.10) and from the richer households 1.54 times higher (AOR = 1.54; 95% CI: 1.19–2.00) than women from the richest households. As the number of total children born before the index child increased by one, the odds of short birth interval increased by 7% (AOR = 1.07; 95% CI: 1.03–1.10). The odds of short birth interval among women whose preceding child did not survive were 1.97 times higher (AOR = 1.97; 95% CI: 1.59–2.45) compared with women whose preceding child survived.

**Community-level factors.** Women residing in a pastoralist region had a two-fold higher odds (AOR = 2.01; 95% CI: 1.68–2.39) and women residing in cities had a 1.75-fold higher odds (AOR = 1.75; 95% CI: 1.38–2.22) of experiencing short birth interval as compared with women residing in an agrarian region. Similarly, women residing in communities with a high proportion of illiterate women had 23% higher odds (AOR = 1.23; 95% CI: 1.05–1.45) of experiencing short birth interval when compared with women residing in communities with a low proportion of illiterate women. Women who lived in a community with increased distance to health facility had a 32% higher odds (AOR = 1.32; 95% CI: 1.11–1.56) of experiencing short birth interval compared with women who lived in a community with close to a health facility.

**Multilevel analysis (random-effects analysis).** Table 4 presents quantities based on random effects. The prevalence rate of short birth interval varied across communities ($\tau^2 = 0.75$, p = <0.001). The empty model revealed that 18.6% of the total variance in the odds of short birth interval was accounted for by between-cluster variation of characteristics (ICC = 0.186). The between-cluster variability declined over successive models, from 18.6% in the empty model to 11.8% in the individual-level only model, 11.2% in the community-level only model, and 9.5% in the final (combined) model. The proportional change in variance indicated that the addition of predictors to the empty model explained an increased proportion of variation in short birth interval. Similar to ICC values, the combined model showed a higher PCV, i.e., 53.3% of variance in

**Table 4. Results from random intercept model (measure of variation) for short birth interval at cluster level using multilevel logistic regression analysis.**

| Random effects (Measure of variation for short birth interval) | Model 1[a] | Model 2[b] | Model 3[c] | Model 4[d] |
|---|---|---|---|---|
| Community-level variance (SE) | 0.75 (0.08) | 0.44 (0.06) | 0.41 (0.05) | 0.35 (0.05) |
| P value | <0.001 | <0.001 | <0.001 | <0.001 |
| ICC (%) | 18.6 | 11.8 | 11.2 | 9.5 |
| Explained variance (PCV) (%) | Reference | 41.3 | 45.3 | 53.3 |
| MOR (95% CI) | 2.28 (2.10, 2.47) | 1.88 (1.74, 2.03) | 1.85 (1.71, 1.98) | 1.75 (1.62, 1.89) |
| Model fit statistics | | | | |
| AIC | 11185.68 | 10503.02 | 10975.21 | 10414.58 |
| BIC | 11199.76 | 10726.92 | 11031.54 | 10680.46 |
| AUC (95% CI) | 0.75 (0.74, 0.76) | 0.74 (0.73, 0.76) | 0.74 (0.73, 0.75) | 0.74 (0.73, 0.75) |

SE = Standard Error; DIC = Deviance Information Criterion; ICC = Intra-Class Correlation; PCV = Percentage Change in Variance; MOR = Median Odds Ratio;

CI = Confidence Interval; AIC = Akaike's Information Criterion; BIC = Schwarz's Bayesian Information Criteria; AUC = Areas under the receiver operating characteristic

Model 1[a] is the null model, a baseline model without any determinant variable

Model 2[b] is adjusted for individual-level factors

Model 3[c] is adjusted for community-level factors

Model 4[d] is the final model adjusted for individual- and community-level factors

short birth interval could be explained by the combined factors at the individual- and community-levels. Furthermore, the MOR confirmed that short birth interval was attributed to community-level factors. The MOR for short birth interval was 2.28 in the empty model, which indicated the presence of variation between communities (clustering) in short birth interval since MOR was two times higher than the reference (MOR = 1). The unexplained community variation in short birth interval decreased to a MOR of 1.75 when all factors were added to the empty model. This implies that even though individual- and community-level factors were considered, the effect of clustering is still statistically significant in the full model.

**Model fit statistics.** As shown in Table 4 (model fit statistics), the values of AIC and BIC showed subsequent reduction which indicates each model represents a significant improvement over the previous model and it points to the goodness of fit of the final model built in the analysis. Therefore, the final (combined) model that incorporated individual- and community-level factors was selected for predicting the occurrence of short birth interval among women. The predictive utility of the final model was fair, with an AUC of 0.74.

## Discussion

The purpose of this paper was to understand the individual- and community-level determinants of short birth interval in a developing country, Ethiopia. Our findings reveal that close to half of women (45.7%) in Ethiopia had a short birth interval. The difference in the prevalence of short birth intervals between the current study and the one reported in the 2016 EDHS (21.7%) [17] is due to the different definitions used for short birth interval. The EDHS considered a birth interval of less than 24 months a short birth interval, whereas our study defined it as less than 33 months, which is in accordance with the WHO recommendations [4]. In the current study, short birth interval births were associated with different individual- and community-level characteristics. At the individual-level, maternal age at first marriage, maternal occupation, husband's/partner's education, husband's/partner's occupation, household wealth index, the total number of children born before the index child, and survival status of the preceding child were associated with short birth interval. At the community-level, women living

in cities and pastoralist regions, in community with a high level of female illiteracy and women who lived far away from a health facility were more likely to experience short birth intervals.

In this study, it was found that the odds of short birth interval among women aged between 20 and 24 years and 25 and 29 years when they first married were higher compared to women who had married at 19 years or under. A previous study conducted in Asian countries found that marriage age influences birth spacing [43]. In contrast, a previous study conducted in Ethiopia reported a non-significant association [19] This inconsistency could be due to the difference in the study setup. Unlike the current study, which was based on nationally representative data, the previous study was restricted to a rural area and single district. Literature reports that age at marriage can influence both the quantum and tempo of fertility [44].

Household wealth index was one of the independent predictors of short birth interval. Women from the poorest, poorer, middle and richer households were more likely to experience short birth interval compared with women from the richest household. This finding is consistent with the existing literature [19]. While it is likely that women with fewer resources have less access to health information and contraception services and thus may not adhere to the scientifically recommended birth intervals, more detailed research is needed to understand this finding.

The study found that death of the preceding child increased the likelihood of short birth interval. It is possible that death of the preceding child ends breastfeeding, resulting in women returning to menses and resuming ovulation sooner; some couples may unintentionally have their next child within a shorter birth interval. On the other hand, some couples may intentionally try to have another child as soon as possible. This finding is consistent with the findings of previous studies performed in Ethiopia [20, 27] and Bangladesh [45].

Our study also found that characteristics of clusters in which women live have an effect on women's experience of short birth interval, independent of their individual characteristics. For instance, residing in a different contextual region such as a pastoralist area and a city affects the odds of women experiencing a short birth interval. Living in a pastoralist region and a city was associated with higher odds of short birth interval. Short birth interval in a pastoralist community could be attributable to their nomadic lifestyle, which hinders access to health services infrastructure, including contraception and media regarding health information. On the other hand, the likelihood of short birth interval among women residing in cities may be explained by the relatively busy lifestyle and expensive cost of living in cities, which may force women to reduce their childbearing period and engage in income-generating activities to earn a living and support their family.

Unlike the effect of individual-level educational status on short birth interval, women living in communities with a high proportion of female illiteracy had 23% higher odds of short birth interval relative to women living in communities with a low proportion of female illiteracy. This result may imply a negative effect of community-level maternal illiteracy on shaping birth interval practice. The effect of community-level female illiteracy may operate through a wide range of mechanisms, such as poor access to information, health illiteracy, family planning unawareness, and contraceptive non-utilization.

The longer the distance to a health facility from the community where the women lived, the higher the odds that women experienced short birth interval. The existing literature [46–48] shows clear connections between distance to a health facility and family planning utilization which is vital to ensuring an optimum birth interval. However, it is important to note that the distance to a health facility in this study was assessed based on women's reports, which may over- or underestimate the actual distance to the health facilities.

The results of our study should be interpreted in light of the following limitations. First, since the information on birth interval was recorded retrospectively, it might be prone to recall

bias. Second, the analyses were conducted using data collected in a cross-sectional survey, which prevents causal inferences. Hence, there is a need for prospective research.

Despite the abovementioned limitations, the study has numerous strengths. First, the dataset used in this analysis was the most recent and nationally representative survey that covered all regions and administrative cities of the country. Second, unlike previously conducted small-scale studies that mainly focused on individual-level factors, the current study considered community-level factors. Additionally, this study applied multilevel modeling to accommodate the hierarchical nature of the EDHS data.

## Conclusion

The study showed that nearly half of women experience short birth interval in Ethiopia. Not only individual characteristics of women, but also community-level factors determine women's experience of short birth interval. Individual-level characteristics, including maternal age at first marriage between 20 and 24, and between 25 and 29, maternal occupation, husband's/partner's education, husband's/partner's occupation, household wealth index, the total number of children born before the index child, and death of the preceding child were found to be independent predictors of short birth interval. Similarly, community-level variables, such as contextual region, community-level female illiteracy, and community-level increased distance to health facility were found to be significantly associated with short birth interval. The random-effects analysis result also revealed that community-level random effects remained significant after controlling for the effects of both individual- and community-level variables, indicating that the experience of short birth interval among women depends on community contexts.

Thus, the prevention of short birth interval requires a multifaceted intervention approach. Family planning programs should pay particular attention to women with the identified risk factors for short birth interval. An approach that may initiate discussion on optimum birth interval is community mobilization through women's groups (e.g., Women's Development Army). The effectiveness of the Women's Development Army in improving maternal and child health services' outcomes has been documented elsewhere [49, 50]. Furthermore, mass campaigns and health information dissemination, particularly in city and pastoralist communities, regarding optimum birth interval is required.

## Implications for policy practice and further research

The government should set a clear target to reduce the magnitude (45.7%) of short birth interval in a defined period. Birth interval messages as per the WHO recommendation should be incorporated into antenatal care, postnatal care, and immunization services to improve women's awareness. Additionally, integrating the WHO recommended birth interval message under the health extension program, which is a community-based health care delivery program [51], could help address women and their husbands/partners in each household. Considering the finding that unemployment affects birth interval, the Ethiopian government should develop initiatives to improve the working status of women and their husbands/partners. Moreover, planners and policymakers should make further efforts to alleviate poverty at the household level. Since community-level illiteracy showed negative externality on women's experience of short birth interval, the Ministry of Health should work in collaboration with the Ministry of Education to improve the literacy status of women in the community. In addition, cultural beliefs related to birth interval practice should be investigated. Policy makers should focus on improving access to health facilities to improve family planning service utilization and reduce short birth interval. Longitudinal data would enable better tracking of birth

interval practice in the country through reducing recall bias and providing data to estimate the causal effect of exposure variables on short birth interval.

## Supporting information

**S1 Table. Description of variables used in the analysis.**
(DOCX)

**S2 Table. The results of multicollinearity analysis.**
(DOCX)

## Acknowledgments

We are grateful to The DHS Program for allowing us to use the EDHS data for this study.

## Author Contributions

**Conceptualization:** Desalegn Markos Shifti, Catherine Chojenta, Elizabeth G. Holliday, Deborah Loxton.

**Data curation:** Desalegn Markos Shifti.

**Formal analysis:** Desalegn Markos Shifti.

**Investigation:** Desalegn Markos Shifti.

**Methodology:** Desalegn Markos Shifti, Catherine Chojenta, Elizabeth G. Holliday, Deborah Loxton.

**Resources:** Desalegn Markos Shifti.

**Software:** Desalegn Markos Shifti.

**Supervision:** Catherine Chojenta, Elizabeth G. Holliday, Deborah Loxton.

**Writing – original draft:** Desalegn Markos Shifti.

**Writing – review & editing:** Desalegn Markos Shifti, Catherine Chojenta, Elizabeth G. Holliday, Deborah Loxton.

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
