## [Decision Letter · Decision Letter 0]

21 Sep 2019

PONE-D-19-21556

Individual and community level determinants of short birth interval in Ethiopia: a multilevel analysis

PLOS ONE

Dear Mr. Shifti,

Thank you for submitting your manuscript to PLOS ONE. After careful consideration, we feel that it has merit but does not fully meet PLOS ONE’s publication criteria as it currently stands. Therefore, we invite you to submit a revised version of the manuscript that addresses the points raised during the review process.

The manuscript has been reviewed by two reviewers and their comments are appended below. The reviewers have raised number of concerns including the lack of theory for conceptualization of the framework of the study, sample inclusion criteria for the analysis and measurement of certain variables among others. I believe that the comments would be helpful to revise the manuscript.   

We would appreciate receiving your revised manuscript by Nov 05 2019 11:59PM. To enhance the reproducibility of your results, we recommend that if applicable you deposit your laboratory protocols in protocols.io, where a protocol can be assigned its own identifier (DOI) such that it can be cited independently in the future. For instructions see: http://journals.plos.org/plosone/s/submission-guidelines#loc-laboratory-protocols

We look forward to receiving your revised manuscript.

Kind regards,

Kannan Navaneetham

Academic Editor

PLOS ONE

Journal Requirements:

Reviewers' comments:

Reviewer's Responses to Questions

**Comments to the Author**

1. Is the manuscript technically sound, and do the data support the conclusions?

Reviewer #1: Yes

Reviewer #2: Yes

2. Has the statistical analysis been performed appropriately and rigorously? 

Reviewer #1: Yes

Reviewer #2: Yes

3. Have the authors made all data underlying the findings in their manuscript fully available?

Reviewer #1: Yes

Reviewer #2: Yes

4. Is the manuscript presented in an intelligible fashion and written in standard English?

Reviewer #1: Yes

Reviewer #2: No

5. Review Comments to the Author

Reviewer #1: The paper by author(s) presents an exhaustive analysis of statistical tools for the analysis of individual and community level determinants of short birth interval in Ethiopia.

This work may be accepted, it is recommended to go through a minor review.

i. Methods: Study area and setting: An in-depth secondary data analysis was conducted using the 2016 Ethiopian Demographic and Health Survey (EDHS) data. and Study design and sampling: the study used data from the 2016 Ethiopian Demographic and Health Survey (EDHS). This may be presents similar meaning.

ii. Why authors used Akaike’s Information Criterion (AIC), Schwarz’s Bayesian information criteria (BIC) and Log-likelihood tests to assess goodness of fit and inform the selection of nested models (individual and community level model)? Why not R2 or adjusted R2 or Mallows Cp or mean sum square error (MSE) or others tools? If authors used any references I think it will be more reliable.

iii. I think authors used fully/mutually adjusted Odds Ratio. If authors describe shortly about this it will be more reliable.

Reviewer #2: Comments on “Individual and community level determinants of short birth interval in Ethiopia: a multilevel analysis” (ID: PONE-D-19-21556)

This paper covers a relevant question: what are the factors associated with short intervals? There is a large body of work examining the effects of short intervals on various outcomes, but we have much less research on the determinants of those intervals in the first place. The paper takes a novel and informative approach to addressing this question by implementing a multilevel mixed effects logit model. I feel that the paper is not ready for publication, as there are a number of issues that need to be addressed.

General comments:

1) The most important overarching weakness of the paper is the lack of any framework to motivate the study and to interpret the results. The authors need to spell out a theoretical argument for how birth intervals are determined. What is the decision-making process? What kinds of factors affect those decisions? How do the major elements included in the analysis (e.g. SES, biological, community, and others) ultimately determine the durations of birth intervals? This will help the authors to interpret their results in some kind of a structured way that can guide the reader. As it is now, the reader is bombarded with many coefficients of varying magnitudes and signs with only brief post hoc arguments justifying them. It would be much more helpful to have an understanding of why we should care about the variables chosen for the model, why the effects are predictable or surprising, and what kind of policy recommendations are called for.

2) The analysis should be restricted to only women with a partner. These make up the great majority of multiparous women (95% in your sample) and these are the individuals which can be realistically targeted with policy interventions. This is because women who have multiple births out of wedlock in a culturally conservative society are unlikely to be planning these births in the same way (if at all). Furthermore, the marital status category (unpartnered) must necessarily be collinear with categories of other variables indicating that the woman is not married. For example, the category for husband’s education indicating ‘not partnered’.

3) The continuous variables (maternal age at marriage) should probably be operationalized in a non-linear way. This can be done either through polynomial transformations (e.g. age at marriage ^2 or higher) or through categorization. Categories are probably a better choice for the presentation of the results. I recommend that the authors examine the data to see if this is appropriate.

4) Does maternal age at birth refer to her age at the birth of the index child or the previous child? If it is her age at the birth of the index child, this variable does not make sense to include in the model. This is because her age at birth is determined by the length of the interval (i.e. your outcome variable). If it is her age at the birth of the previous child, you need to spell this out in the text. Likewise, children ever born should refer to the number of children born prior to the birth of the index child (i.e. not including the index child). It was not clear from the paper or supporting documentation.

5) If I have understood the results correctly, it seems like community-level factors actually have very little influence on birth interval length once individual-level characteristics are controlled for. The ICC for the full model with individual and community factors was only 2.9%. In the conclusion and policy implications, however, you have interpreted community-level factors as a major concern. Is this justified given the low ICC?

6) The paper will benefit greatly from language editing for clarity and precision.

Specific comments:

Line 46: The definition of birth intervals is not quite right. First, the interval is not defined as the number of months between births, but the amount of time. Months are simply the unit that some authors express that time in. It can equivalently be expressed in years (in continuous form) or days. Second, the authors are defining a preceding birth interval here. That is fine, because that is what the paper is about. I would simply suggest that they rephrase the definition as the definition of a preceding interval instead of birth interval in general. This is because there can also be a succeeding birth interval, which would be the interval following the birth of the index child.

Line 70: How can the prevalence range from 21% to 57.6%? Does this mean across specific sub-populations or over time?

Line 223: Age at birth should be reported in more meaningful categories than the ones provided here. The age group 18-34 is enormous and captures the large majority of births. Why not present these as standard 5-year age groups?

Line 228: ‘Magnitude’ should really be ‘prevalence’.

Line 302 – 314: If you are using the same data as the EDHS, I do not think there is a need to explain why your estimates of the prevalence of short intervals differ from the report. It is clear that you use a different definition of ‘short intervals’ than the DHS program, and that this is the reason why your prevalence is higher. The comparisons to Tanzania, DR Congo, and Bangladesh are not really relevant here. I would recommend dropping this paragraph entirely, and earlier in the paper when you present the prevalence you estimate, just add a line stating that the definition you use is different from the DHS definition, and this is why your estimates are higher.

Table 1: Age at first marriage is not included in the table.

Table 3: Where is model 1? The columns start from model 2. I would also set the reference category of education as ‘No education’. Only 2.5% of women had higher education.

6. PLOS authors have the option to publish the peer review history of their article (what does this mean?). If published, this will include your full peer review and any attached files.

Reviewer #1: Yes: Benojir Ahammed

Reviewer #2: No

---

## [Author Response · Author response to Decision Letter 0]

14 Nov 2019

Editorial comments: When submitting your revision, we need you to address these additional requirements. Please ensure that your manuscript meets PLOS ONE's style requirements, including those for file naming. The PLOS ONE style templates can be found at http://www.journals.plos.org/plosone/s/file?id=wjVg/PLOSOne_formatting_sample_main_body.pdf and http://www.journals.plos.org/plosone/s/file?id=ba62/PLOSOne_formatting_sample_title_authors_affiliations.pdf

Response: Our manuscript has been revised to meet PLOS ONE's style requirements.

Reviewer #1 comments: The paper by author(s) presents an exhaustive analysis of statistical tools for the analysis of individual and community level determinants of short birth interval in Ethiopia.

Response: Thank you.

Reviewer #1 comments: Methods: Study area and setting: An in-depth secondary data analysis was conducted using the 2016 Ethiopian Demographic and Health Survey (EDHS) data. and Study design and sampling: the study used data from the 2016 Ethiopian Demographic and Health Survey (EDHS). This may be presents similar meaning.

Response: We agree with the reviewer and we removed the repetition from the ‘study design and sampling’ section.

Reviewer #1 comments: Why authors used Akaike’s Information Criterion (AIC), Schwarz’s Bayesian information criteria (BIC) and Log-likelihood tests to assess goodness of fit and inform the selection of nested models (individual and community level model)? Why not R2 or adjusted R2 or Mallows Cp or mean sum square error (MSE) or others tools? If authors used any references, I think it will be more reliable.

Response: Thank you. Generally, R-squared (R2), Mean Squared Error (MSE) and other tools (such as Residual Standard Error (RSE), Mean Absolute Error (MAE)) are sensitive to the inclusion of additional variables in the model, even if those variables don’t explain significant variation in the outcome. For instance, including additional variables in the model will always increase the R2 and reduce the MSE. In contrast, Akaike’s Information Criterion (AIC), and Schwarz’s Bayesian information criteria (BIC) penalise the deviance for models with a larger number of parameters, and thus provide more protection against overfitting the model to the data, relative to approaches based on hypothesis testing (e.g., deviance difference, log-likelihood ratio statistic). Therefore, using a more robust metric to guide the model choice is recommended. Adjusted R2, AIC, BIC, and Mallows Cp are among the most commonly used metrics for measuring regression model quality and for comparing models. It is because of the above-mentioned reasons, we used AIC and BIC. In this revised version, we removed Log-likelihood tests. Therefore, in this revised version of the manuscript, we restricted the model fit statistics based on Akaike’s Information Criteria (AIC) and Schwarz’s Bayesian Information Criteria (BIC). We have added a brief statement explaining the robust nature of AIC and BIC under the ‘model fit statistics’ section of the manuscript (Line 230-236).

Reviewer #1 comments: I think authors used fully/mutually adjusted Odds Ratio. If authors describe shortly about this it will be more reliable.

Response: We agree and have mentioned that ‘adjusted odds ratio’ was used to express the fixed effect sizes of the individual- and community-level factors on short birth interval (Line 197)

Reviewer #2 comments: This paper covers a relevant question: what are the factors associated with short intervals? There is a large body of work examining the effects of short intervals on various outcomes, but we have much less research on the determinants of those intervals in the first place. The paper takes a novel and informative approach to addressing this question by implementing a multilevel mixed effects logit model. I feel that the paper is not ready for publication, as there are a number of issues that need to be addressed.

Response: We have carefully edited the manuscript to address the issues raised by you and other reviewers

Reviewer #2 comments: The most important overarching weakness of the paper is the lack of any framework to motivate the study and to interpret the results. The authors need to spell out a theoretical argument for how birth intervals are determined. What is the decision-making process? What kinds of factors affect those decisions? How do the major elements included in the analysis (e.g. SES, biological, community, and others) ultimately determine the durations of birth intervals? This will help the authors to interpret their results in some kind of a structured way that can guide the reader. As it is now, the reader is bombarded with many coefficients of varying magnitudes and signs with only brief post hoc arguments justifying them. It would be much more helpful to have an understanding of why we should care about the variables chosen for the model, why the effects are predictable or surprising, and what kind of policy recommendations are called for.

Response: The socio-ecological model of health behavior provides the theoretical foundation of the current study. We have now incorporated detail about this theoretical foundation into the manuscript (line 126-139). In addition, as mentioned in the method section (line 165-170), the assumption of independence of observation has been taken as a basis to determine which variables are to be analysed at the individual- and community-levels.

Reviewer #2 comments: The analysis should be restricted to only women with a partner. These make up the great majority of multiparous women (95% in your sample) and these are the individuals which can be realistically targeted with policy interventions. This is because women who have multiple births out of wedlock in a culturally conservative society are unlikely to be planning these births in the same way (if at all). Furthermore, the marital status category (unpartnered) must necessarily be collinear with categories of other variables indicating that the woman is not married. For example, the category for husband’s education indicating ‘not partnered’.

Response: The variable ‘Marital status’ in Demographic and Health Survey data describes the marital status of the women at the time of the survey not their marital status in the last five years. In our previous version of the manuscript, the ‘not married’ category included women who had never been married, as well as women who were separated, divorced, and widowed. In light of your feedback, in this revised version, ‘marital status’ is now categorized into ‘married’ and ‘separated/divorced/widowed.

With the exception of ‘never married’ women, we assumed that all other unpartnered women (i.e. separated, divorced, or widowed) who were eligible for the birth interval question at the time of the survey conceived their multiple births from with their late/ex-husband. Since data regarding birth interval were collected from their childbirths in the five years preceding the survey, women may become widowed or divorced or separated shortly (or any time) before the time of the survey but after they already become eligible for providing birth interval information. Therefore, separated, divorced, and widowed women were retained for the analysis. In addition, while separated, divorced, and widowed women were with their late/ex-husbands, there were likely factors present that limited them from achieving an optimum birth interval and these factors should be investigated. Moreover, since some of separated/divorced/widowed women were among those who contributed to the magnitude of short birth interval in the country, including them in the analysis could help to inform policy and health programmers. However, as you hypothesized that ‘women who have multiple births out of wedlock in a culturally conservative society may be unlikely to be planning their births in the same way’, we excluded ‘never married’ (i.e. 12 (0.14%)) women from the analysis in this revised version of the manuscript. Thank you for your careful review. We have described the exclusion of ‘never-married women’ from the analysis (line 115 to 117). In addition, the reasons for retaining separated/divorced/widowed women are described under the exposure variable section (line 147-148). Regarding your concern regarding collinearity among ‘Not partnered’ categories and not married women:

In our previous version of the manuscript, since separated, divorced, and widowed women (i.e. 371 women) were not eligible to answer for their polygyny status, their husband/partner’s education, and their husband's/partner’s occupation, there was systematic missingness for these variables. To address the missing values, we replaced them with a ‘Not partnered’ category values. However, your concern regarding collinearity led us to consider the variable as it is (i.e. with the systematic missingness). Thank you for that. Now, we performed a complete case analysis (CCA). Evidence has shown that CCA provides unbiased results when the chance of being a complete case does not depend on the outcome (i.e. the chance of being complete case for the variable husbands’ education, husbands’ occupation, and polygyny status does not depend on whether the women experienced short birth interval or not) (Hughes RA, Heron J, Sterne JA, Tilling K. Accounting for missing data in statistical analyses: multiple imputation is not always the answer. International journal of epidemiology. 2019;1:11.). Therefore, we believe the concern of collinearity has been addressed. In addition, the results of multicollinearity analysis are annexed as supplementary material (S2 Table. The results of multicollinearity analysis. (DOCX)).

Reviewer #2 comments: The continuous variables (maternal age at marriage) should probably be operationalized in a non-linear way. This can be done either through polynomial transformations (e.g. age at marriage ^2 or higher) or through categorization. Categories are probably a better choice for the presentation of the results. I recommend that the authors examine the data to see if this is appropriate.

Response: We agree with the reviewer and have presented ‘maternal age at first marriage’ in 5-year age categories. However, the highest category is restricted to maternal age at marriage of 30 years and above since the number of observations after 30 years of maternal age becomes small.

Reviewer #2 comments: Does maternal age at birth refer to her age at the birth of the index child or the previous child? If it is her age at the birth of the index child, this variable does not make sense to include in the model. This is because her age at birth is determined by the length of the interval (i.e. your outcome variable). If it is her age at the birth of the previous child, you need to spell this out in the text. Likewise, children ever born should refer to the number of children born prior to the birth of the index child (i.e. not including the index child). It was not clear from the paper or supporting documentation.

Response: Thank you. Previously, we considered maternal age at the birth of the index child. In this revised manuscript, we used maternal age at the birth of the preceding child. Since it does not fulfill the linearity assumption, we presented it in 5-year age categories.

Reviewer #2 comments: If I have understood the results correctly, it seems like community-level factors actually have very little influence on birth interval length once individual-level characteristics are controlled for. The ICC for the full model with individual and community factors was only 2.9%. In the conclusion and policy implications, however, you have interpreted community-level factors as a major concern. Is this justified given the low ICC?

Response: We appreciate your observation. However, an ICC value of 2.9% while small, it is not zero. It indicates that the experience of short birth interval among women still depends on community contexts despite controlling individual- and community-level factors in the final model. 

Moreover, in this revised of the manuscript, the magnitude of ICC in the final model increased (from 2.9% to 9.5%). This change could be attributed to the changes made in coding some of exposure variables such as maternal age at first marriage, maternal age at birth of preceding child and total number of children born before the index child. Keeping this change in mind, even though individual- and community-level factors were controlled in the final model, 9.5% of the total variance in the odds of short birth interval was accounted for by between-cluster variation of characteristics. This implies that experience of short birth interval among women still depends on community contexts. As it is known, an ICC of 0 or closer to 0 would suggest that the clusters/community are similar to random samples taken from the country and suggests that clusters/community are not relevant to understand the variation in short birth interval. Whereas a high ICC value informs us that clusters/community are very important in understanding the individual differences in short birth interval (Merlo J, Chaix B, Yang M, Lynch J, Råstam L. A brief conceptual tutorial of multilevel analysis in social epidemiology: linking the statistical concept of clustering to the idea of contextual phenomenon. Journal of Epidemiology & Community Health. 2005;59(6):443-9.). In line with this, our interest is to show that nearly 10% of the total individual differences in short birth interval were due to cluster/community variation. In the manuscript (line 404-407), the statement was presented as follow: 

“The random-effect analysis result had also revealed that community-level random effects remained significant after controlling for both individual- and community-level variables, indicating that the experience of short birth interval among women depends on community contexts.”

Reviewer #2 comments: The paper will benefit greatly from language editing for clarity and precision.

Response: The paper has been carefully revised by native English speakers.

Reviewer #2 comments: Line 46: The definition of birth intervals is not quite right. First, the interval is not defined as the number of months between births, but the amount of time. Months are simply the unit that some authors express that time in. It can equivalently be expressed in years (in continuous form) or days. Second, the authors are defining a preceding birth interval here. That is fine, because that is what the paper is about. I would simply suggest that they rephrase the definition as the definition of a preceding interval instead of birth interval in general. This is because there can also be a succeeding birth interval, which would be the interval following the birth of the index child.

Response: The definition has been corrected accordingly. 

Reviewer #2 comments: Line 70: How can the prevalence range from 21% to 57.6%? Does this mean across specific sub-populations or over time?

Response: The variation of the magnitude was observed across sub-populations. The 21% prevalence of short birth interval is based on the 2016 Ethiopia Demographic and Health Survey while 57.6% prevalence was based on the 2010 study conducted in one district; Lemo district. For clarity, it is rewritten as follows (line 74-75): 

‘In Ethiopia, the prevalence of short birth interval ranges from 21.0% (national estimate) (17) to 57.6% (district level estimate) (18).’

Reviewer #2 comments: Line 223: Age at birth should be reported in more meaningful categories than the ones provided here. The age group 18-34 is enormous and captures the large majority of births. Why not present these as standard 5-year age groups?

Response: Age at birth is presented in the 5-year age categories as per your recommendation.

Reviewer #2 comments: Line 228: ‘Magnitude’ should really be ‘prevalence’.

Response: Corrected accordingly.

Reviewer #2 comments: Line 302 – 314: If you are using the same data as the EDHS, I do not think there is a need to explain why your estimates of the prevalence of short intervals differ from the report. It is clear that you use a different definition of ‘short intervals’ than the DHS program, and that this is the reason why your prevalence is higher. The comparisons to Tanzania, DR Congo, and Bangladesh are not really relevant here. I would recommend dropping this paragraph entirely, and earlier in the paper when you present the prevalence you estimate, just add a line stating that the definition you use is different from the DHS definition, and this is why your estimates are higher.

Response: This paragraph is entirely dropped from the paper. The reason for the difference is also described as follow (line 325-329):

‘The difference in the prevalence of short birth intervals between the current study and the one reported in the 2016 EDHS (21.7%) (17) is due to the different definitions used for short birth interval. The EDHS considered a birth interval of less than 24 months a short birth interval, whereas our study defined it as less than 33 months, which is in accordance with the WHO recommendations (4).’

Reviewer #2 comments: Table 1: Age at first marriage is not included in the table.

Response: It is incorporated in this revised manuscript.

Reviewer #2 comments: Table 3: Where is model 1? The columns start from model 2. 

Response: As it is mentioned in the method section, under the heading ‘multivariable multilevel analysis’ (Line 190-193), Model I (Empty model) was fitted without explanatory variables to estimate random variation in the intercept and the intra-cluster correlation coefficient (ICC). Table 3 presents quantities based on fixed effects only. That is why column describing model 1 was not included in table 3. The finding of model 1 is presented in table 4. However, for the sake of clarity, the reason for not including ‘Model I’ in table 3 has been added as a footnote (line 608).

Reviewer #2 comments: I would also set the reference category of education as ‘No education’. Only 2.5% of women had higher education.

Response: The reference category of education has been changed as per your suggestion.

---

## [Decision Letter · Decision Letter 1]

31 Dec 2019

Individual and community level determinants of short birth interval in Ethiopia: a multilevel analysis

PONE-D-19-21556R1

Dear Dr. Shifti,

We are pleased to inform you that your manuscript has been judged scientifically suitable for publication and will be formally accepted for publication once it complies with all outstanding technical requirements.

With kind regards,

Kannan Navaneetham

Academic Editor

PLOS ONE

Additional Editor Comments (optional):

Reviewers' comments:

Reviewer's Responses to Questions

**Comments to the Author**

1. If the authors have adequately addressed your comments raised in a previous round of review and you feel that this manuscript is now acceptable for publication, you may indicate that here to bypass the “Comments to the Author” section, enter your conflict of interest statement in the “Confidential to Editor” section, and submit your "Accept" recommendation.

Reviewer #1: All comments have been addressed

2. Is the manuscript technically sound, and do the data support the conclusions?

Reviewer #1: Yes

3. Has the statistical analysis been performed appropriately and rigorously? 

Reviewer #1: Yes

4. Have the authors made all data underlying the findings in their manuscript fully available?

Reviewer #1: Yes

5. Is the manuscript presented in an intelligible fashion and written in standard English?

Reviewer #1: Yes

6. Review Comments to the Author

Reviewer #1: (No Response)

7. PLOS authors have the option to publish the peer review history of their article (what does this mean?). If published, this will include your full peer review and any attached files.

Reviewer #1: No

---

## [Editor Report · Acceptance letter]

6 Jan 2020

PONE-D-19-21556R1 

Individual and community level determinants of short birth interval in Ethiopia: a multilevel analysis 

Dear Dr. Shifti:

I am pleased to inform you that your manuscript has been deemed suitable for publication in PLOS ONE. Congratulations! Your manuscript is now with our production department. 

With kind regards,

on behalf of

Professor Kannan Navaneetham 

Academic Editor

PLOS ONE